# SGMem: Sentence Graph Memory for Long-Term Conversational Agents

## Abstract

Long-term conversational agents require effective memory management to handle dialogue histories that exceed the context window of large language models (LLMs). Existing methods based on fact extraction or summarization reduce redundancy but struggle to organize and retrieve relevant information across different granularities of dialogue and generated memory. We introduce SGMem (Sentence Graph Memory), which represents dialogue as sentence-level graphs within chunked units, capturing associations across turn-, round-, and session-level contexts. By combining retrieved raw dialogue with generated memory such as summaries, facts and insights, SGMem supplies LLMs with coherent and relevant context for response generation. Experiments on LongMemEval and LoCoMo show that SGMem consistently improves accuracy and outperforms strong baselines in long-term conversational question answering.

## 1 Introduction

Memory is a fundamental component of long-term conversational agents (Maharana et al., 2024; Wu et al., 2024), allowing them to augment dialogue context beyond the limited window of large language models (LLMs) (Zhang et al., 2025a; Wu et al., 2025; Sapkota et al., 2025). By acquiring, storing, managing, and retrieving information from prior interactions, memory supports accurate and personalized responses in multi-turn conversations. However, as interactions accumulate, agents inevitably face *memory overload* (Klingberg, 2009; Yun et al., 2010), where the scale, complexity, or redundancy of stored content exceeds their ability to manage and retrieve it effectively. This condition undermines dialogue understanding and constrains the agent's capacity to deliver coherent and user-tailored responses.

Memory management (Xiong et al., 2025; Kang et al., 2025) seeks to organize, compress, and filter stored content to improve an agent's ability to exploit large-scale memory and alleviate *memory overload*. In long-term conversational agents, memory typically consists of both the *raw dialogue history*—spanning turns, rounds, and sessions—and *generated memory* such as summaries, extracted facts, and reflective insights. While techniques like summarization, extraction, and reflection reduce redundancy, they often lead to the so-called *memory fragmentation* (Bedard-Gilligan & Zoellner, 2012; Kindt & Van Den Hout, 2003), where relevant information is dispersed across raw dialogues and derived snippets, hindering coherent retrieval. Figure 1 provides an overview of memory in long-term conversational agents for question answering (QA), which is typically formulated within a retrieval-augmented generation (RAG) to retrieve relevant memory segments and reduce hallucination (Siriwardhana et al., 2023; Fan et al., 2024).

Despite efforts from existing chunk-based and graph-based approaches (Wu et al., 2025; Pan et al., 2025; Zhang et al., 2025b)—such as employing memory composition and updating strategies inspired by Zettelkasten note-taking (Kadavy, 2021; Ahrens, 2022), or modeling entity–relation associations via event-centric memory graphs (Zhang et al., 2025b)—the problem of memory fragmentation remains largely unresolved. On the one hand, it is still challenging to determine the appropriate granularity at which raw dialogue history should be retrieved and to effectively integrate generated memory with raw history during retrieval. On the other hand, extracting entity-relation triples with LLMs incurs substantial computational costs and further exacerbates memory fragmentation.

To address these challenges, our central design choice is to structure dialogue memory at the sentence level. Sentences serve as the fundamental units of conversational exchange, each encapsulating a

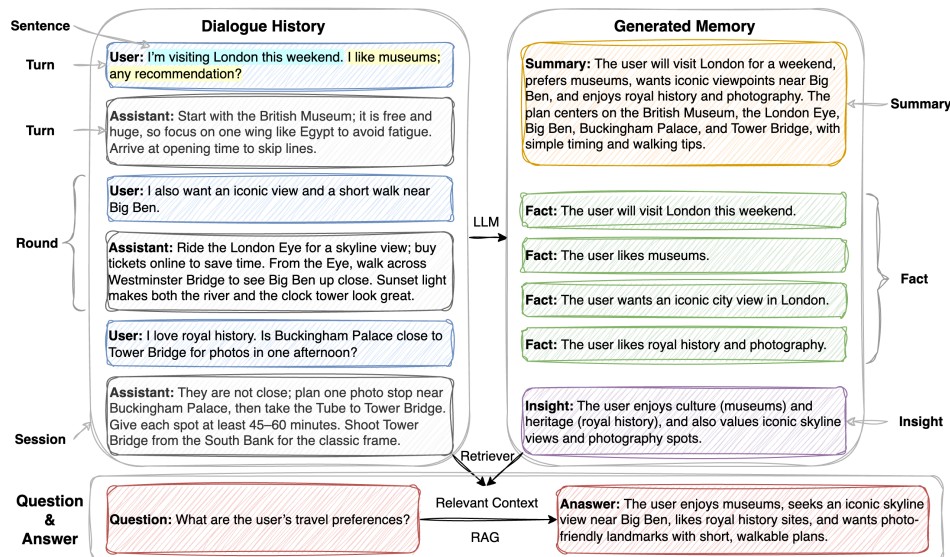

Figure 1: Illustration of memory in long-term conversational agents.

semantically coherent statement while remaining fine-grained enough to capture contextual dependencies. Compared with coarser units (e.g., turns, rounds, sessions), sentence-level representations enable more precise alignment between raw dialogue history and generated memory. Moreover, structuring sentences as nodes in a graph allows the agent to explicitly model associations—both within and across dialogue segments—thereby mitigating memory fragmentation and supporting coherent retrieval.

In this paper, we propose SGMem (Sentence Graph Memory), a memory management framework that improves the organization and retrieval of long-term conversational memory. SGMem models dialogue as sentence-level graphs within chunked units, enabling associations across turns, rounds, and sessions. By jointly retrieving raw dialogue history and generated memory (e.g., summaries, facts, and insights), SGMem supplies LLMs with coherent and contextually relevant evidence for response generation. Notably, SGMem requires no additional LLM-based extraction; it relies only on standard sentence segmentation tools (e.g., NLTK (Bird, 2006)) to construct sentence graphs, making it lightweight and readily deployable in long multi-turn conversational settings.

The contributions of this paper are threefold:

• **Sentence-Graph Memory Construction & Management**: We introduce SGMem, a hierarchical memory framework that mitigates memory fragmentation by organizing dialogue history into sentence-level graphs.

• **Sentence-Graph Memory Usage**: We design a multi-hop retrieval mechanism over the sentence graph memory that integrates raw dialogue history with generated memory to support coherent and accurate long-term question answering.

• **Comprehensive Evaluation**: We conduct extensive experiments on LongMemEval and LoCoMo, showing that SGMem achieves consistent accuracy gains and outperforms strong baselines, demonstrating both effectiveness and practical value.

## 2 RELATED WORK

**Long-Term Conversational Agents.** Long-term conversational agents are designed to sustain multi-session interactions, but face challenges such as memory retention and update, temporal reasoning, context granularity, and coherent retrieval over fragmented histories. To benchmark these abilities, recent work has introduced dedicated datasets. LongMemEval (Wu et al., 2024) evaluates personal assistants on five memory skills—information extraction, multi-session reasoning, temporal reasoning, knowledge updates, and abstention—revealing significant performance drops in multi-session and temporally dynamic settings. LoCoMo (Maharana et al., 2024) provides very long persona-grounded, event-driven conversations spanning up to 35 sessions, with tasks such as question answering, event

summarization, and multimodal dialogue generation. Results on both benchmarks show that existing LLMs and RAG pipelines struggle with temporal consistency, knowledge updates, and coherent retrieval, underscoring the need for more structured and fine-grained memory management.

**Memory Management.** Memory management is a central challenge for long-term conversational agents (Wu et al., 2025; Xu et al., 2025a). Existing methods include MemoryBank (Zhong et al., 2024), which hierarchically summarizes events and aggregates personality insights; LD-Agent (Li et al., 2024), which separates long- and short-term memory banks for event summaries and contextual dialogue while updating user personas; RMM (Tan et al., 2025), which reflects on dialogue to form topic-based summaries; MemoryScope (Yu et al., 2024), which consolidates user observations into higher-level insights; and A-MEM (Xu et al., 2025b), which draws on Zettelkasten (Kadavy, 2021; Ahrens, 2022) to link memories as structured notes. Despite these advances, memory fragmentation—where information is scattered across raw dialogue and generated memory—remains unresolved. This motivates our approach, SGMem, which represents dialogue at the sentence level as a graph to align fine-grained semantics and support coherent retrieval.

**Retrieval Augmented Generation.** Retrieval-augmented generation (RAG) is a dominant paradigm for grounding LLMs with external knowledge in long-term conversational agents (Jin et al., 2024). Chunk-based RAG retrieves dialogue segments at the turn, round, or session level, offering simplicity and scalability but often limited by coarse granularity and fragmented retrieval. In contrast, graph-based RAG methods (Zhang et al., 2025a; Han et al., 2024), such as GraphRAG (Edge et al., 2024), LightRAG (Guo et al., 2024), and HippoRAG (Jimenez Gutierrez et al., 2024; Gutiérrez et al., 2025), construct structured indexes over entities, relations, or hierarchical clusters to capture richer cross-document associations. In line with this perspective, graph-based RAG has been employed to construct memory graphs centered on entity relationships, exemplified by the event-centric memory graph (Zhang et al., 2025b). A key limitation of entity-level memory graphs is their reliance on costly LLM computations for entity and relation extraction, which also discards rich contextual information. To address this issue, we introduce SGMem, a lightweight sentence-level graph memory that eliminates the need for LLM-based extraction while retaining the semantic content of sentences.

## 3 METHODOLOGY

### 3.1 PRELIMINARIES

We consider the task of long-term conversational question answering (QA), where the input consists of a sequence of *sessions* denoted as $\mathcal{S} = \{s_1, s_2, \ldots, s_U\}$. Each session $s_u$ contains multiple *turns* $\mathcal{T} = \{t_w\}_{w=1}^W$, which can be further grouped into user-assistant *rounds* $\mathcal{R} = \{r_v\}_{v=1}^V$ to reflect higher-level conversational exchanges. In addition to raw sessions, long-term conversational agents often maintain various forms of *generated memory*, including *summaries* $\mathcal{M} = \{m_x\}_{x=1}^X$, *facts* $\mathcal{F} = \{f_y\}_{y=1}^Y$, and *insights* $\mathcal{I} = \{i_z\}_{z=1}^Z$, produced by large language models (LLMs) through summarization, extraction, or reflection. At a finer granularity, each turn $t_w$ can be segmented into a set of *sentences* $\mathcal{C} = \{c_j\}_{j=1}^J$ using standard NLP tools such as NLTK (Bird, 2006). These hierarchical units—sessions, rounds, turns, generated memory, and sentences—form the basis of our Sentence Graph Memory (SGMem) management and retrieval framework.

### 3.2 FRAMEWORK OVERVIEW

Long-term conversational agents often suffer from coarse memory segmentation, where both raw dialogue history (turns, rounds, sessions) and generated memories (summaries, facts, insights) are stored and retrieved at coarse granularity, leading to fragmented and incoherent context. To address this limitation, we propose **Sentence Graph Memory (SGMem)**, which organizes dialogue at the sentence level and explicitly models semantic associations through graph structures. Figure 2 presents an overview of SGMem, which consists of two main components: (1) *SGMem Construction & Management*, and (2) *SGMem Usage*.

**SGMem Construction & Management.** The construction of Sentence Graph Memory (SGMem) consists of four steps: (1) *Processing Conversations*: Sessions are hierarchically decomposed into rounds, turns, and sentences, while LLMs generate summaries, facts, and insights. (2) *Indexing*: All memory units are embedded into vector spaces to build seven searchable index tables. (3)

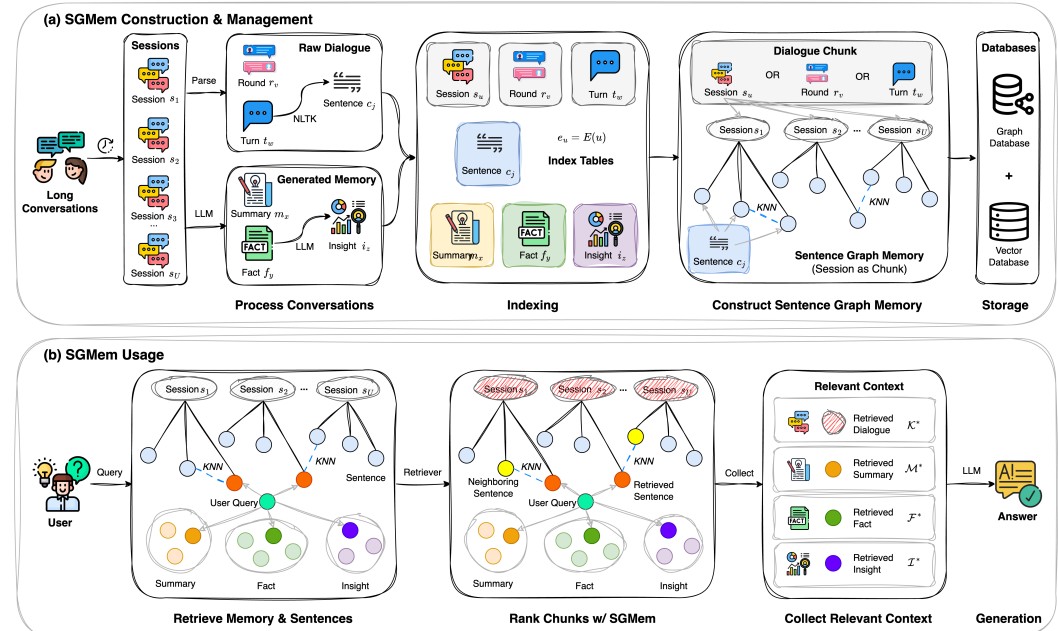

Figure 2: Overview of the proposed **Sentence Graph Memory (SGMem)** framework, consisting of (a) *SGMem Construction & Management* for building sentence-level memory graphs and (b) *SGMem Usage* for retrieving relevant memory in long-term conversational QA.

*Constructing Sentence Graph Memory*: Chunk nodes (sessions, rounds, or turns) are linked to their constituent sentences and further connected by sentence–sentence similarity edges. (4) *Storage*: Index tables are stored in a vector database for efficient search, and the sentence graph memory is maintained in a graph database for reasoning and traversal. This design yields a structured and queryable memory foundation for SGMem Usage.

**SGMem Usage.** The usage of Sentence Graph Memory (SGMem) consists of four steps. (1) *Retrieve Memory and Sentences*: The query retrieves candidate summaries, facts, insights, and sentences from the vector database. (2) *Rank Chunks with SGMem*: Retrieved sentences are expanded via $n$-hop graph traversal and mapped back to their parent chunks, which are ranked and truncated for relevance. (3) *Collect Relevant Context*: Selected chunks together with generated memories are aggregated into a unified relevant context. (4) *Personalized Generation*: The aggregated context is fed into the LLM to produce accurate and personalized responses. Overall, this dual design of vector retrieval and graph expansion ensures coherent context selection for long-term conversational QA.

### 3.3 SGMEM CONSTRUCTION & MANAGEMENT

Sentence Graph Memory (SGMem) is constructed and managed through four steps, forming a structured and queryable foundation for its usage.

**Processing Conversations.** Given a set of long conversations $\mathcal{S} = \{s_u\}_{u=1}^{U}$, we decompose each session $s_u$ into *rounds* $\mathcal{R} = \{r_v\}_{v=1}^{V}$ and *turns* $\mathcal{T} = \{t_w\}_{w=1}^{W}$. Each turn $t_w$ is further segmented into a set of *sentences* $\mathcal{C} = \{c_j\}_{j=1}^{J}$ using standard NLP tools (e.g., NLTK (Bird, 2006)). In parallel, we employ an LLM to derive three types of generated memory:

$$\mathcal{M} = \{m_x\}_{x=1}^{X}, \quad \mathcal{F} = \{f_y\}_{y=1}^{Y}, \quad \mathcal{I} = \{i_z\}_{z=1}^{Z},$$

where $\mathcal{M}, \mathcal{F}, \mathcal{I}$ denote summaries, facts, and insights, respectively.

**Indexing.** Each memory unit $u \in \{s, r, t, c, m, f, i\}$ is encoded into a vector representation $\mathbf{e}_u \in \mathbb{R}^d$ using a pretrained embedding model $E(\cdot)$, such as Sentence-BERT (Reimers & Gurevych, 2019): $\mathbf{e}_u = E(u)$. This produces seven index tables corresponding to sessions, rounds, turns, sentences, summaries, facts, and insights.

**Constructing Sentence Graph Memory.** Raw dialogue units (session, round, or turn) are treated as *chunk* nodes $\mathcal{K} = \{k_p\}$, and each chunk $k_p$ is linked to its constituent sentences $c_j$ through membership edges:

$$(k_p, c_j) \in \mathcal{E}_{\text{chunk-sent}} \quad \text{if } c_j \in k_p.$$

In addition, we compute pairwise sentence similarity: $\text{sim}(c_j, c_{j'}) = \cos(\mathbf{e}_{c_j}, \mathbf{e}_{c_{j'}})$, and construct a $k$-nearest-neighbor (KNN) graph:

$$(c_j, c_{j'}) \in \mathcal{E}_{\text{sent-sent}} \quad \text{if } c_{j'} \in \text{Top-}k \text{ neighbors of } c_j.$$

The overall sentence graph memory is thus defined as:

$$\mathcal{G} = (\mathcal{V}, \mathcal{E}_{\text{chunk-sent}} \cup \mathcal{E}_{\text{sent-sent}}), \quad \text{where} \quad \mathcal{V} = \mathcal{K} \cup \mathcal{C}.$$

**Storage.** The seven index tables $\{\mathbf{e}_u\}$ are stored in a vector database (e.g., ElasticSearch[1]) for efficient similarity search, while the graph $\mathcal{G}$ is stored in a graph database (e.g., Neo4j[2]) to support reasoning and traversal-based retrieval.

## 3.4 SGMEM USAGE

The usage of Sentence Graph Memory (SGMem) consists of four steps designed to maximize retrieval coherence and personalization.

**Retrieve Memory and Sentences.** Given a query $q$, we retrieve candidate summaries $\mathcal{M}$, facts $\mathcal{F}$, insights $\mathcal{I}$, and sentences $\mathcal{C}$ from their respective index tables in the vector database. Retrieval is based on cosine similarity:

$$\text{sim}(q, u) = \cos(\mathbf{e}_q, \mathbf{e}_u) + \epsilon, \quad u \in \{m_x, f_y, i_z, c_j\},$$

where $\mathbf{e}_q$ and $\mathbf{e}_u$ are embeddings of query $q$ and memory unit $u$, and $\epsilon = 1$. The top-$K$ memory units are retained as $\mathcal{M}^*$, $\mathcal{F}^*$, and $\mathcal{I}^*$, respectively. A similarity threshold $\gamma \in [0, 2]$ and a maximum number of sentence nodes $n$ are employed to constrain the retrieval process of sentence nodes.

**Rank Chunks with SGMem.** Retrieved sentences $\mathcal{C}_q$ are expanded via $h$-hop traversal on the sentence graph $\mathcal{G}$ to gather neighbors $\mathcal{N}_h(\mathcal{C}_q)$:

$$\mathcal{C}^* = \mathcal{C}_q \cup \mathcal{N}_h(\mathcal{C}_q).$$

Each sentence $c_j \in \mathcal{C}^*$ is mapped back to its parent chunk $k_p$ (session, round, or turn), and chunks are ranked by an aggregate score:

$$score(k_p) = \frac{1}{|\mathcal{C}_{k_p}|} \sum_{c_j \in \mathcal{C}_{k_p}} \text{sim}(q, c_j),$$

where $\mathcal{C}_{k_p}$ is the set of retrieved and neighboring sentences belonging to chunk $k_p$. Top-$K$ chunks are retained as $\mathcal{K}^*$.

**Collect Relevant Context.** The final relevant context is the union of retrieved chunks, summaries, facts, and insights:

$$\mathcal{C}_{\text{relevant}} = \mathcal{K}^* \cup \mathcal{M}^* \cup \mathcal{F}^* \cup \mathcal{I}^*.$$

**Personalized Generation.** The LLM generates an output response $\hat{y}$ conditioned on the query $q$ and the relevant context $\mathcal{C}_{\text{relevant}}$:

$$\hat{y} = \text{LLM}(q \,|\, \mathcal{C}_{\text{relevant}}).$$

Overall, SGMem Usage benefits from the dual design of vector-based retrieval and graph-based expansion: vector indexes provide efficient access to heterogeneous memory types, while sentence graph traversal ensures contextual coherence, leading to improved response accuracy and personalization in long-term conversational QA.

---

[1] https://github.com/elastic/elasticsearch
[2] https://github.com/neo4j/neo4j

## 4 EXPERIMENTAL SETTINGS

To comprehensively evaluate SGMem, we aim to answer the following research questions: • **RQ1:** Does SGMem outperform existing memory management and RAG methods in long-term conversational QA? • **RQ2:** How do different types of context (e.g., raw dialogue units vs. generated memory) influence QA effectiveness? • **RQ3:** How does SGMem perform across different query types? • **RQ4:** How do the hyperparameters (e.g., $k$, $h$, $n$, $\gamma$) affect the performance of SGMem?

### 4.1 DATASETS

We conduct experiments on two long-term conversational benchmarks. **LongMemEval** (Wu et al., 2024) comprises 500 curated questions spanning six types—single-session-user (70), single-session-assistant (56), single-session-preference (30), multi-session (133), knowledge-update (78), and temporal-reasoning (133)—embedded in user–assistant dialogues of varying length. **LoCoMo** (Maharana et al., 2024) provides very long multi-session dialogues (300 turns, 9K tokens, up to 35 sessions) grounded in personas and temporal event graphs; we randomly sample 500 questions, covering single-hop (156), multi-hop (133), temporal reasoning (133), and open-domain knowledge (78), to ensure computational feasibility and enable extensive ablation studies. Together, these datasets jointly evaluate fine-grained memory abilities and scalability to very long, multi-session interactions. The dataset details are provided in Appendix A.

### 4.2 EVALUATION METRIC

We evaluate long-term conversational question answering using **Accuracy**, where correctness of a model response is determined by the **LLM-as-a-Judge** paradigm (Gu et al., 2024). Specifically, a strong LLM is prompted to compare the generated response against the reference answer and decide whether it is correct. This design avoids the brittleness of exact string matching, allowing the metric to account for paraphrases and semantically equivalent answers while still providing a clear accuracy score. We report accuracy across different question types in both datasets. To ensure reproducibility, we adopt a fixed evaluation prompt and provide the full prompt template in the Appendix B.

### 4.3 BASELINES

**Simple Baselines.** **No History**: Answers questions without considering dialogue history, using only the query itself as input. **Long Context**: Directly feeds the LLM with dialogue history, either the most recent sessions ($LC_{Latest}$) or all sessions ($LC_{Full}$).

**Memory Management Baselines.** **MemoryBank** (Zhong et al., 2024): Maintains chronological memory with hierarchical summaries and uses *rounds + summaries* as context. **LD-Agent** (Li et al., 2024): Disentangles long- and short-term memory banks and uses *summaries + facts* as context. **LongMemEval** (Wu et al., 2024): Builds indexes over sessions augmented with corresponding facts, retrieving *sessions* as context. **MemoryScope** (Yu et al., 2024): Performs consolidation and reflection, using *rounds + facts + insights* as context. **RMM** (Tan et al., 2025): Applies prospective reflection over history and uses *facts* as context.

**Graph-based RAG Baselines.** **LightRAG** (Guo et al., 2024): Constructs lightweight relational graphs and uses *entities + relations* as context. **MiniRAG** (Fan et al., 2025): Compresses conversational memory into smaller graph structures and uses *sessions + entities* as context. **KG-Retriever** (Chen et al., 2024): Builds hierarchical knowledge graphs and uses *relations* as context.

**Chunk-based RAG Variants.** We also implement chunk-based RAG variants by varying the memory unit used as retrieval context. Specifically, we evaluate RAG with *turns* (RAG-T, RAG-TF, RAG-TMFI), *rounds* (RAG-R, RAG-RF, RAG-RMFI), and *sessions* (RAG-S, RAG-SF, RAG-SMFI), where "TF" denotes turns with *facts*, and "TMFI" denotes turns with *summaries, facts, and insights*. For each variant, we retrieve the top-$K$ items from the specified memory types (e.g., turns, summaries, facts, insights) and concatenate them as context. SGMem is evaluated under the same variants for fair comparison.

### 4.4 SETUP

**Retriever.** We use Sentence-BERT (Reimers & Gurevych, 2019) for dense retrieval, specifically the `all-MiniLM-L6-v2` model for embedding sentences and memory units. It is also employed

| Method | Mode | Graph | T | R | S | M | F | I | E | L | LongMemEval Top-5 | LongMemEval Top-10 | LoCoMo Top-5 | LoCoMo Top-10 |
|---|---|---|---|---|---|---|---|---|---|---|---|---|---|---|
| No History | LLM | - | ✗ | ✗ | ✗ | ✗ | ✗ | ✗ | ✗ | ✗ | 0.000 | 0.000 | 0.050 | 0.050 |
| $LC_{Latest}$ | LLM | - | ✗ | ✗ | ✓ | ✗ | ✗ | ✗ | ✗ | ✗ | 0.144 | 0.196 | 0.196 | 0.292 |
| $LC_{Full}$ | LLM | - | ✗ | ✗ | ✓ | ✗ | ✗ | ✗ | ✗ | ✗ | 0.478 | 0.478 | 0.558* | 0.558* |
| MemoryBank | RAG | - | ✗ | ✓ | ✗ | ✓ | ✗ | ✗ | ✗ | ✗ | 0.498 | 0.558 | 0.388 | 0.422 |
| LD-Agent | RAG | - | ✗ | ✗ | ✗ | ✓ | ✓ | ✗ | ✗ | ✗ | 0.502 | 0.574 | 0.418 | 0.434 |
| LongMemEval | RAG | - | ✗ | ✗ | ✓ | ✗ | ✗ | ✗ | ✗ | ✗ | 0.552 | 0.556 | 0.346 | 0.410 |
| MemoryScope | RAG | - | ✗ | ✓ | ✗ | ✗ | ✓ | ✓ | ✗ | ✗ | 0.642 | 0.678 | 0.430 | 0.468 |
| RMM | RAG | - | ✗ | ✗ | ✗ | ✗ | ✓ | ✗ | ✗ | ✗ | 0.612 | 0.668 | - | - |
| LightRAG | RAG | KG | ✗ | ✗ | ✗ | ✗ | ✗ | ✗ | ✓ | ✓ | 0.420 | 0.428 | 0.360 | 0.406 |
| MiniRAG | RAG | KG | ✗ | ✗ | ✓ | ✗ | ✗ | ✗ | ✓ | ✗ | 0.422 | 0.468 | 0.268 | 0.336 |
| KG-Retriever | RAG | HIG | ✗ | ✗ | ✗ | ✗ | ✗ | ✗ | ✗ | ✓ | 0.112 | 0.104 | 0.138 | 0.124 |
| RAG-T | RAG | - | ✓ | ✗ | ✗ | ✗ | ✗ | ✗ | ✗ | ✗ | 0.456 | 0.544 | 0.286 | 0.330 |
| RAG-R | RAG | - | ✗ | ✓ | ✗ | ✗ | ✗ | ✗ | ✗ | ✗ | 0.478 | 0.564 | 0.284 | 0.352 |
| RAG-S | RAG | - | ✗ | ✗ | ✓ | ✗ | ✗ | ✗ | ✗ | ✗ | 0.574 | 0.576 | 0.340 | 0.408 |
| RAG-SF | RAG | - | ✗ | ✗ | ✓ | ✗ | ✓ | ✗ | ✗ | ✗ | 0.656 | 0.684 | 0.478 | 0.502 |
| RAG-SMFI | RAG | - | ✗ | ✗ | ✓ | ✓ | ✓ | ✓ | ✗ | ✗ | 0.676 | 0.680 | 0.510 | 0.528 |
| SGMem-S | RAG | SG | ✗ | ✗ | ✓ | ✗ | ✗ | ✗ | ✗ | ✗ | 0.644 | 0.614 | 0.392 | 0.476 |
| SGMem-SF | RAG | SG | ✗ | ✗ | ✓ | ✗ | ✓ | ✗ | ✗ | ✗ | 0.690 | **0.730*** | 0.522 | **0.542** |
| SGMem-SMFI | RAG | SG | ✗ | ✗ | ✓ | ✓ | ✓ | ✓ | ✗ | ✗ | **0.700*** | **0.730*** | **0.526** | 0.532 |

Table 1: Performance comparison on LongMemEval and LoCoMo using Accuracy (Top-5 / Top-10). KG denotes knowledge graph, HIG denotes hierarchical index graph, SG denotes sentence graph. T, R, S, M, F, I, E, L denote turn, round, session, summary, fact, insight, entity, and relation respectively. The overall best results are marked with *, the best RAG method is shown in **bold**, and the second-best RAG method is underlined.

to compute sentence similarities when constructing the $k$-nearest-neighbor (KNN) graph in SGMem, where BM25 (Robertson et al., 2009) is adopted for a fair comparison.

**LLM.** For both question answering and evaluation tasks, we employ a state-of-the-art instruction-tuned model with 32B parameters, `Qwen2.5-32B-Instruct` (Qwen et al., 2025). LLM was accessed via the BaiLian[3] API platform with default generation parameters: $temperature = 0.7$, $top\_p = 0.8$, $top\_k = 20$, $max\_input\_tokens = 129{,}024$, and $max\_tokens = 8{,}192$.

**Hyperparameters.** By default, for **LongMemEval**, we set $k = 3$, $h = 1$, $n = 15$, $\gamma = 1.0$, and $K = 5$, with ablation studies conducted on the `SGMem-TF` variant (Turn + Fact). For **LoCoMo**, we use $k = 1$, $h = 1$, $n = 15$, $\gamma = 1.2$, and $K = 5$, with ablations conducted on the `SGMem-SF` variant (Session + Fact). The ranges explored in hyperparameter sensitivity analysis are: $k \in \{1, 2, 3, 4, 5\}$, $h \in \{0, 1, 2\}$, $n \in \{5, 10, 15, 20\}$, $\gamma \in \{1.0, 1.2, 1.5\}$, $K \in \{5, 10\}$.

## 5 EXPERIMENTAL RESULTS

Extensive experiments are performed to evaluate SGMem against strong baselines (Section 5.1), analyze the effect of different context types (Section 5.2), investigate its performance across different query types (Section 5.3), and study the sensitivity of hyperparameters (Section 5.4). This section addresses four research questions (RQ1–RQ4 in Section 4).

### 5.1 SGMEM VS. BASELINES

To address RQ1, we compare SGMem against a broad set of representative baselines (Section 4.3). Table 1 reports results on LongMemEval and LoCoMo using Accuracy (Top-5 / Top-10). Among the simple baselines, directly using no history yields near-zero performance, while $LC_{Latest}$ and $LC_{Full}$ show limited improvement on LongMemEval, highlighting the insufficiency of naive long-context usage. For memory management baselines, methods such as MemoryBank, LD-Agent, and LongMemEval achieve moderate gains, whereas more advanced strategies like MemoryScope and RMM substantially improve accuracy by introducing structured summaries, facts, and insights. Graph-based approaches, including LightRAG, MiniRAG, and KG-Retriever, underperform on

---

[3]https://bailian.console.aliyun.com/

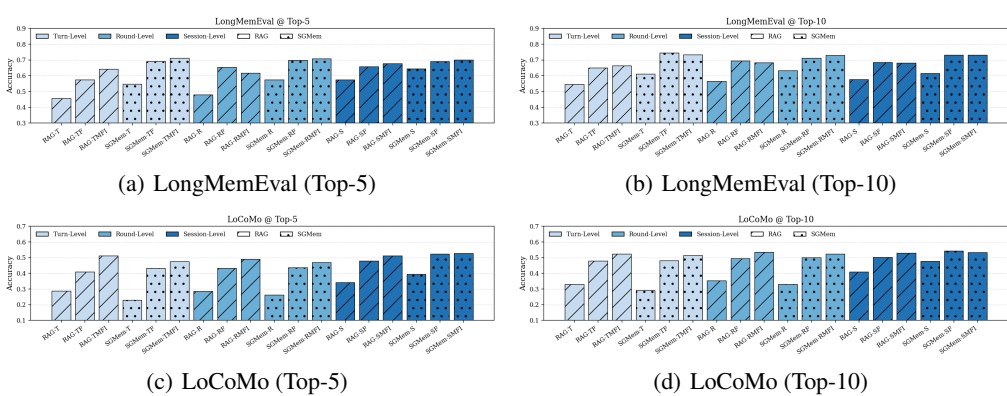

Figure 3: Performance comparison of RAG and SGMem variants on LongMemEval and LoCoMo under Top-5 and Top-10 settings. Turn-, round-, and session-level denote raw dialogue units; M, F, I denote summary, fact, and insight, respectively.

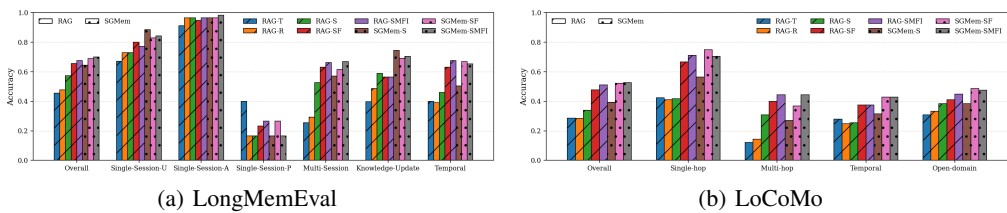

Figure 4: QA performance across various Query subsets on both datasets.

both benchmarks, due to their reliance on entity- or relation-level graphs that lack alignment with conversational granularity. RAG variants that retrieve turns, rounds, or sessions provide stronger results, with RAG-SMFI (sessions + summaries, facts, insights) achieving the best performance among RAG-based baselines (0.676/0.680 on LongMemEval and 0.510/0.528 on LoCoMo).

In contrast, SGMem consistently outperforms all RAG-based baselines. SGMem-SF achieves 0.690/0.730 on LongMemEval and 0.522/0.542 on LoCoMo, while SGMem-SMFI further improves to 0.700 and 0.526 at Top-5, respectively—the best RAG-based results across both datasets. These improvements demonstrate the advantage of representing dialogue at the sentence level and explicitly modeling associations via graph structures, which mitigates memory fragmentation and enables more coherent retrieval. Overall, in response to RQ1, SGMem consistently outperforms existing memory management and RAG baselines on both LongMemEval and LoCoMo, establishing a new paradigm for long-term conversational QA.

### 5.2 IMPACT OF CONTEXT TYPE

To addresss RQ2, we analyze variants of SGMem and RAG that use turns, rounds, sessions, summaries, facts, and insights as context, and investigate the trade-off between fidelity (raw history) and conciseness (generated memory). Figure 3 compares RAG and SGMem variants under different context settings. We observe that using only raw dialogue units (turns, rounds, sessions) provides a faithful but fragmented context, often yielding limited gains. For instance, turn-level RAG (RAG-T) performs the weakest, while session-level RAG (RAG-S) achieves stronger accuracy, showing the importance of larger dialogue spans. In contrast, incorporating generated memory—summaries, facts, and insights—substantially improves performance across all granularities. Variants such as RAG-SF and RAG-SMFI outperform their raw-only counterparts, confirming that generated memory enhances relevance and reduces noise. SGMem further amplifies these benefits by leveraging sentence-level graphs: SGMem-SF and SGMem-SMFI consistently surpass corresponding RAG variants, indicating that sentence-level associations help align raw dialogue with generated memory. Overall, in response to RQ2, the results highlight that combining raw dialogue with generated memory is essential for effective retrieval, and that SGMem provides a principled way to integrate both.

### 5.3 PERFORMANCE ACROSS QUERY TYPES

Figure 4 presents accuracy results across query subsets for both LongMemEval and LoCoMo. SGMem consistently outperforms RAG variants across all query types, confirming its robustness

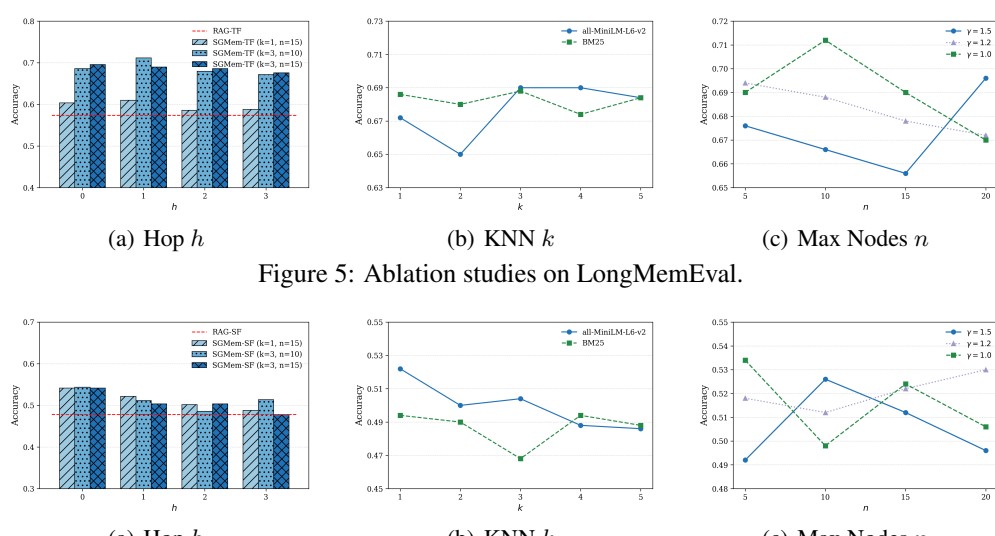

Figure 5: Ablation studies on LongMemEval.

(a) Hop $h$      (b) KNN $k$      (c) Max Nodes $n$

Figure 6: Ablation studies on LoCoMo.

under diverse conversational scenarios. On LongMemEval, we find that the largest improvements appear in *multi-session*, *knowledge-update*, and *temporal reasoning* queries, where effective reasoning requires bridging fragmented or evolving user information. Single-session queries (user and assistant) are relatively easier, where all methods achieve higher performance, but SGMem still maintains a margin over RAG baselines. On LoCoMo, SGMem also shows clear advantages across *single-hop*, *temporal*, and *open-domain* queries, highlighting its ability to capture both fine-grained details and long-range dependencies. Overall, in response to RQ3, these results demonstrate that sentence-graph memory provides consistent benefits across query types, particularly in settings that demand temporal tracking, multi-session integration, and adaptation to evolving knowledge.

## 5.4 IMPACT OF HYPERPARAMETERS

Figures 5 and 6 report ablation studies on LongMemEval and LoCoMo by varying hop $h$, KNN size $k$, maximum nodes $n$, and weighting factor $\gamma$. The results highlight that SGMem's performance depends on careful tuning, and its optimal configurations differ across datasets. On LongMemEval, moderate values of $h$ and $k$ (e.g., $h = 1$, $k = 3$) yield small gains, while extreme settings bring diminishing returns. Accuracy peaks around $n = 10$, and $\gamma = 1.0$ provides slightly stronger results than larger scaling factors. Between retrievers for KNN, BM25 remains more stable across various $k$, although the dense retriever (`all-MiniLM-L6-v2`) achieves stronger peak accuracy when tuned properly. On LoCoMo, increasing $h$ consistently degrades accuracy, and larger $k$ or $n$ often introduce noise. Compared with LongMemEval, LoCoMo benefits more from careful tuning, with $\gamma = 1.2$ producing relatively stable results. BM25 again demonstrates robustness under variation, though at the cost of slightly lower best-case accuracy compared to the dense retriever. Overall, in response to RQ4, these findings suggest that SGMem requires hyperparameter calibration to achieve optimal performance, with LongMemEval being more tolerant to variation, while LoCoMo demands more careful tuning due to its longer and noisier conversational histories.

## 6 CONCLUSION

Long-term conversational agents demand robust memory management to overcome the limitations of LLM context windows and support accurate, personalized responses. In this paper, we introduced SGMem, a sentence graph memory framework that organizes dialogue into sentence-level graphs, bridging raw dialogue histories and generated memory through explicit associations. By integrating turns, rounds, and sessions with summaries, facts, and insights, SGMem provides coherent and contextually grounded evidence for response generation. Experiments on LongMemEval and LoCoMo demonstrate that SGMem consistently outperforms strong baselines, yielding improvements across diverse query types and showing robustness to hyperparameter variations. These results highlight the effectiveness of sentence-level graph memory in mitigating fragmentation and redundancy, paving the way toward more scalable and reliable long-term conversational agents.

## LIMITATIONS

While SGMem demonstrates consistent improvements over strong baselines, several limitations remain. First, although SGMem effectively integrates raw dialogue and generated memory, it does not yet address hallucinations or factual inconsistencies that may arise from LLM-generated summaries, facts, or insights. Second, our evaluation is conducted on two benchmarks (LongMemEval and LoCoMo), which, despite their coverage of diverse query types, may not fully capture the breadth of real-world conversational dynamics such as multimodal contexts, streaming updates, or highly personalized long-term memory. Finally, SGMem has not been optimized for efficiency at scale; constructing and maintaining sentence-level graphs over very large histories may incur additional computational and storage overhead. Future work could explore fact-verification mechanisms, multimodal extensions, and scalable graph maintenance to further enhance the reliability and applicability of SGMem.

## ETHICS STATEMENT

This work introduces SGMem, a sentence-level graph memory framework for long-term conversational agents. Our experiments are conducted exclusively on publicly available datasets that do not contain personally identifiable information or sensitive content. The proposed SGMem aims to improve accuracy in memory management without altering or fabricating raw dialogue content. While our approach does not involve direct human subjects, we note that summaries, facts, and insights are generated using large language models (LLMs), which may introduce biases or inaccuracies inherent to the models. We therefore encourage practitioners to apply SGMem responsibly, ensuring that both the raw dialogue histories and LLM-generated content are handled with appropriate safeguards to protect user privacy, mitigate bias amplification, and prevent misuse in deployment.

## REPRODUCIBILITY STATEMENT

We have made careful efforts to ensure the reproducibility of SGMem. All datasets used in our experiments are publicly available and described in detail in Section 4 and Appendix A. The construction of sentence-level graph memory, retrieval configurations, evaluation metrics, and prompting strategies are fully specified in the paper and appendix. Hyperparameters for all SGMem variants are reported. Collectively, these details should enable independent researchers to replicate our results without reliance on proprietary resources.

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

## A  DATASET STATISTICS

**LongMemEval.**  LongMemEval (Wu et al., 2024) is a large-scale benchmark designed to evaluate five core memory abilities of LLM-based chat assistants: information extraction, multi-session reasoning, temporal reasoning, knowledge updates, and abstention. It contains **500 curated questions** embedded in user–assistant dialogues of varying length and complexity. Each question is annotated with its type and aligned with corresponding sessions that provide supporting evidence. The distribution of question types includes: 70 *single-session-user*, 56 *single-session-assistant*, 30 *single-session-preference*, 133 *multi-session*, 78 *knowledge-update*, and 133 *temporal-reasoning*. For example, a *single-session-user* question is:

```
Question:  What degree did I graduate with?
Answer:  Business Administration
Question_Type:  single-session-user
Question_Date:  2023/05/30 (Tue) 23:40
Evidence:  Session ID [answer_280352e9]
```

This benchmark presents a challenging setting where existing long-context LLMs and commercial chat assistants show significant accuracy degradation when information must be recalled across extended interactions.

**LoCoMo.**  LoCoMo (Maharana et al., 2024) is a very long-term conversational benchmark generated via a machine–human pipeline that grounds multi-session dialogues on personas and temporal event graphs. Each conversation averages 300 turns and 9K tokens across up to 35 sessions, and some dialogues incorporate multimodal interactions (e.g., image sharing and reactions). To ensure computational feasibility, we randomly sample **500 questions** from the full set of 1,986 annotated questions. These questions are distributed across four categories: 156 *single-hop*, 133 *multi-hop*, 133 *temporal reasoning*, and 78 *open-domain knowledge*. For example, a *temporal reasoning* question is:

```
Question:  Which country was Jolene located in during the
last week of August 2023?
Answer:  Brazil
Question_Type:  temporal reasoning
Evidence:  Dialogue ID [D23:1]
```

Experiments on LoCoMo highlight the difficulty of modeling long-range temporal and causal dynamics, where long-context LLMs and RAG systems still lag behind human performance.

## B  PROMPTS

To facilitate reproducibility, we provide the full set of prompts used in our experiments. These include the RESPONSE PROMPT (Appendix B.1), EVALUATION PROMPT (Appendix B.2), SUMMARY PROMPT (Appendix B.3), FACT PROMPT (Appendix B.4), and INSIGHT PROMPT (Appendix B.5), which are designed for response generation, model evaluation, dialogue summarization, fact extraction, and insight generation, respectively.

### B.1  RESPONSE PROMPT

**Prompt B.1: Response Prompt**

```
---Role---
```

```
You are a helpful assistant responding to questions about data
    provided.

---Goal---

Generate a response of the target length and format that responds to
    the user's question, summarizing all information in the input data
    appropriate for the response length and format, and incorporating
    any relevant general knowledge.
If you don't know the answer, just say so. Do not make anything up.
Do not include information where the supporting evidence for it is not
     provided.

---Target response length and format---

Multiple Paragraphs

Add sections and commentary to the response as appropriate for the
    length and format. Style the response in markdown.
```

## B.2 EVALUATION PROMPT

**Prompt B.2: Evaluation Prompt**

```
---Role---

You are a helpful evaluation assistant.
You will be given a question, a gold-standard answer, and a candidate
    answer generated via retrieval-augmented generation (RAG).

---Goal---

Evaluate the candidate answer against the gold-standard answer based
    on factual accuracy and completeness in answering the question.

Scoring Criteria:
- score=1 (Correct): The candidate answer is factually accurate and
    fully or reasonably paraphrases the gold-standard answer.
- score=0 (Incorrect): The candidate answer is factually incorrect,
    irrelevant, incomplete, or does not answer the question.

---Output Format---

Provide your evaluation in the following JSON format:

```json
{
  "score": X
}
```
where X is either 1 or 0.
```

## B.3 SUMMARY PROMPT

**Prompt B.3: Summary Prompt**

```
---Role---

You are a helpful summarization assistant.

---Goal---

Please summarize the following dialogue as concisely as possible,
    extracting the main themes and key information. If there are
    multiple key events, you may summarize them separately.
```

## B.4 FACT PROMPT

**Prompt B.4: Fact Prompt**

```
--- Role ---

You are a precise and helpful fact extraction assistant.
You will be given a list of conversation messages between a human user
     and an AI assistant.

--- Goal ---

Extract **all explicit personal facts** about the human user,
    including but not limited to:
- Life events (past, present, or planned)
- Personal experiences
- Preferences and interests
- Relationships and interactions with people
- Numbers, dates, locations, organizations, and other concrete details

Each extracted fact must:
1. Be a **standalone, self-contained sentence**.
2. Avoid pronouns (replace "I", "my", "she", "they" with explicit
    entities, e.g., "The user", "Maya", "Jake Watson").
3. Preserve all available details (time, place, quantity, frequency,
    etc.).
4. Remain strictly factual (do not infer, summarize, or speculate
    beyond the given text).

If no personal facts are found, output an empty list.

--- Output Format ---

Return the facts as a JSON list of strings, where each string is one
    fact:

```json
["fact 1", "fact 2", "fact 3"]
```
```

## B.5 INSIGHT PROMPT

**Prompt B.5: Insight Prompt**

```
--- Role ---

You are a precise and helpful fact reflection assistant.
You will be given a list of factual records about a human user.

--- Goal ---

Your task is to analyze the provided user memories and generate higher
    -level, insightful reflections.

--- Analysis and Reflection Rules ---

- Carefully read the memory entries and identify recurring themes,
    behaviors, or connections.
- Multiple related memory entries may be **merged into a single
    insight** if they collectively represent a pattern or habit (e.g.,
    repeated actions --> user habit).
- The 'timestamp' of the generated insight should be the **latest
    timestamp** among the merged memory entries.
- Reflect and summarize to generate higher-level insights such as user
     preferences, habits, routines, opinions, goals, or current status.
- Each insight should be concise, self-contained, and written as a
    standalone statement.
- Do not copy the input facts directly; instead, abstract them into
    meaningful patterns or insights.
- Do not include any explanations, metadata, or comments outside of
    the JSON output.

--- Input Format ---

A list of user memories, each containing a timestamp and content:

[
  {'timestamp': '', 'content': ''},
  {'timestamp': '', 'content': ''},
  {'timestamp': '', 'content': ''},
  ...
]

--- Output Format ---

Return the insights as a **JSON list of objects**.
Each object must have:
- 'timestamp': the latest timestamp among the related memory entries
- 'content': the generated insight

```json
[
  {'timestamp': '', 'content': ''},
  {'timestamp': '', 'content': ''},
  ...
]
```
```

## C CASE STUDIES

We provide case studies on both LongMemEval (Appendix C.1) and LoCoMo (Appendix C.2) to compare RAG-SMFI with SGMem-SMFI. The examples demonstrate that RAG-SMFI often fails

to answer questions correctly based on retrieved memory alone, whereas SGMem-SMFI leverages multi-hop traversal within SGMem to locate relevant chunks (i.e., session-level information) and complement them with generated memory, leading to correct answers.

## C.1 A CASE STUDY ON LONGMEMEVAL

**RAG-SMFI on LongMemEval.** We present an illustrative case of long-term conversational QA evaluation with RAG-SMFI on LongMemEval, as shown in Use Case C.1. In this example, RAG-SMFI fails to provide the correct answer.

---

**Use Case C.1: RAG-SMFI on LongMemEval**

```
{
"question_id": "0bc8ad92",
"question_type": "temporal-reasoning",
"question_date": "2023/03/25 (Sat) 17:18",
"question": "How many months have passed since I last visited a museum
    with a friend?",
"answer": 5,
"answer_session_ids": ["answer_f4ea84fb_3", "answer_f4ea84fb_2", "
    answer_f4ea84fb_1"],
"results": {
"response": "Based on the information provided, your last visit to a
    museum with a friend occurred on a recent date, specifically a
    behind-the-scenes tour of the Science Museum with a chemistry
    professor friend. This visit was referenced in the context of the
    user's recent museum visits, which included a guided tour at the
    Natural History Museum on February 18, 2023.\n\nSince the timestamp
     for the Natural History Museum visit is February 18, 2023, and the
     current timestamp is March 25, 2023, approximately one month has
    passed since your last mentioned museum visit.\n\nHowever, if you
    specifically visited the Science Museum with your chemistry
    professor friend after the Natural History Museum visit, please
    clarify the exact date of that visit. If no specific date was given
     for the Science Museum visit, we can assume it was within the same
     timeframe, meaning about one month has passed since your last
    museum visit."},
"score": 0
}
```

---

**SGMem-SMFI on LongMemEval.** Use Case C.2 illustrates a long-term conversational QA evaluation with our proposed SGMem-SMFI on LongMemEval, where the model provides the correct answer.

---

**Use Case C.2: SGMem-SMFI on LongMemEval**

```
{
"question_id": "0bc8ad92",
"question_type": "temporal-reasoning",
"question_date": "2023/03/25 (Sat) 17:18",
"question": "How many months have passed since I last visited a museum
    with a friend?",
"answer": 5,
"answer_session_ids": ["answer_f4ea84fb_3", "answer_f4ea84fb_2", "
    answer_f4ea84fb_1"],
"results": {
"topk_sentence_ids": ["answer_f4ea84fb_1_9_12", "answer_f4ea84fb_3_5_0
    ", "answer_f4ea84fb_3_5_14", "answer_f4ea84fb_1_11_10", "
```

---

```
    answer_f4ea84fb_3_1_4", "answer_f4ea84fb_3_5_15", "
    answer_f4ea84fb_1_8_0", "answer_f4ea84fb_3_1_3", "
    answer_f4ea84fb_3_4_0", "answer_f4ea84fb_3_7_4", "
    answer_f4ea84fb_3_0_0", "answer_f4ea84fb_1_9_0", "
    answer_f4ea84fb_1_10_1", "answer_f4ea84fb_3_7_2", "
    answer_f4ea84fb_3_6_3"],
 "chunk_to_sentences": {"answer_f4ea84fb_3": ["answer_f4ea84fb_3_1_5",
    "answer_f4ea84fb_3_8_0", "answer_f4ea84fb_3_1_4", "
    answer_f4ea84fb_3_7_2", "answer_f4ea84fb_3_7_4", "
    answer_f4ea84fb_3_1_3", "answer_f4ea84fb_3_5_0", "
    answer_f4ea84fb_3_7_10", "answer_f4ea84fb_3_5_15", "
    answer_f4ea84fb_3_0_0", "answer_f4ea84fb_3_6_3", "
    answer_f4ea84fb_3_6_5", "answer_f4ea84fb_3_1_10", "
    answer_f4ea84fb_3_7_7", "answer_f4ea84fb_3_5_14", "
    answer_f4ea84fb_3_4_0", "answer_f4ea84fb_3_7_5"], "
    answer_f4ea84fb_1": ["answer_f4ea84fb_1_10_1", "
    answer_f4ea84fb_1_9_13", "answer_f4ea84fb_1_9_12", "
    answer_f4ea84fb_1_9_0", "answer_f4ea84fb_1_10_2", "
    answer_f4ea84fb_1_8_0", "answer_f4ea84fb_1_1_1", "
    answer_f4ea84fb_1_11_10"]},
 "topk_chunk_ids": ["answer_f4ea84fb_1", "answer_f4ea84fb_3"],
 "response": "Based on the information provided, your last visit to a
    museum with a friend occurred on October 22, 2022. Given the
    current timestamp of March 25, 2023, approximately five months have
     passed since your last museum visit with your friend.\n\nTo break
    it down:\n- Last museum visit with a friend: October 22, 2022\n-
    Current date: March 25, 2023\n\nThat means it has been roughly five
     months since your last museum visit with your chemistry professor
    friend. If you're considering another visit soon, you might want to
     check the museums' schedules and any new exhibits that may have
    opened in the meantime."},
 "score": 1
 }
```

## C.2 A Case Study on LoCoMo

**RAG-SMFI on LoCoMo.** Use Case C.3 illustrates a long-term conversational QA evaluation with RAG-SMFI on LoCoMo, where the model fails to provide the correct answer.

**Use Case C.3: RAG-SMFI on LoCoMo**

```
{
 "question_id": "conv-47",
 "question_type": "temporal reasoning",
 "question": "When did James depart for his trip to Canada?",
 "answer": "July 11, 2022",
 "answer_dialogue_ids": ["D16:9"],
 "answer_session_ids": ["conv-47_s15"],
 "results": {
 "response": "Based on the information provided, James bought air
    tickets to Toronto and planned to visit Vancouver as well. However,
     the exact departure date for his trip to Canada is not specified
    in the timestamps provided. The only dates mentioned are related to
     his plans to return from the trip on July 20 and a discussion
    about his travel plans taking place at 5:13 pm on July 9, 2022.
    Therefore, we can infer that James likely departed sometime after
    July 9, 2022, but before July 20, 2022, the date he intended to
    return. Without additional information, the precise departure date
    cannot be determined."},
```

```
"score": 0}
```

**SGMem-SMFI on LoCoMo.** As shown in Use Case C.4, our proposed SGMem-SMFI successfully answers the long-term conversational QA task on LoCoMo.

**Use Case C.4: SGMem-SMFI on LoCoMo**

```
{
"question_id": "conv-47",
"question_type": "temporal reasoning",
"question": "When did James depart for his trip to Canada?",
"answer": "July 11, 2022",
"answer_dialogue_ids": ["D16:9"],
"answer_session_ids": ["conv-47_s15"],
"results": {
"topk_sentence_ids": ["conv-47_s15_10_0", "conv-47_s6_9_0", "conv-47
    _s5_13_0", "conv-47_s29_2_0", "conv-47_s5_11_0", "conv-47_s30_24_0
    ", "conv-47_s15_6_0", "conv-47_s27_34_0", "conv-47_s19_21_0", "conv
    -47_s16_35_0", "conv-47_s20_10_0", "conv-47_s5_15_0", "conv-47
    _s27_20_0", "conv-47_s16_15_0", "conv-47_s12_9_0"],
"chunk_to_sentences": {"conv-47_s13": ["conv-47_s13_0_0"], "conv-47
    _s22": ["conv-47_s22_20_0"], "conv-47_s10": ["conv-47_s10_17_0"], "
    conv-47_s19": ["conv-47_s19_21_0"], "conv-47_s27": ["conv-47
    _s27_24_0", "conv-47_s27_34_0", "conv-47_s27_20_0"], "conv-47_s20":
     ["conv-47_s20_10_0"], "conv-47_s16": ["conv-47_s16_15_0", "conv-47
    _s16_35_0", "conv-47_s16_23_0"], "conv-47_s15": ["conv-47_s15_6_0",
     "conv-47_s15_10_0"], "conv-47_s29": ["conv-47_s29_2_0"], "conv-47
    _s12": ["conv-47_s12_9_0"], "conv-47_s6": ["conv-47_s6_9_0"], "conv
    -47_s8": ["conv-47_s8_21_0"], "conv-47_s5": ["conv-47_s5_15_0", "
    conv-47_s5_11_0", "conv-47_s5_17_0", "conv-47_s5_13_0"], "conv-47
    _s17": ["conv-47_s17_15_0"], "conv-47_s30": ["conv-47_s30_24_0"], "
    conv-47_s0": ["conv-47_s0_11_0"], "conv-47_s26": ["conv-47_s26_13_0
    "], "conv-47_s11": ["conv-47_s11_6_0"]},
"topk_chunk_ids": ["conv-47_s6", "conv-47_s15", "conv-47_s29", "conv
    -47_s5", "conv-47_s30"],
"response": "Based on the information provided, James departed for his
     trip to Canada the day after the conversation he had with John on
     July 9, 2022. Since their conversation took place at 5:13 pm on
    July 9, 2022, and James mentioned that he was leaving \"the day
    after tomorrow evening,\" his departure would have been on July 11,
     2022, in the evening. His planned return date from the trip was
    July 20, 2022."},
"score": 1
}
```

