# OpenReview forum: "SGMem: Sentence Graph Memory for Long-Term Conversational Agents"
_ICLR.cc/2026/Conference — Submitted to ICLR 2026_

### Official Review · Reviewer_o6ii · 2025-10-29

**Soundness:** 2
**Presentation:** 3
**Contribution:** 2
**Rating:** 2
**Confidence:** 4

**Summary:**

This paper proposes SGMem (Sentence Graph Memory), a framework for managing long-term conversational memory beyond LLM context limits. It represents dialogues as sentence-level graphs to organize information across turns, rounds, and sessions. By integrating retrieved dialogues with generated summaries, facts, and insights, SGMem provides coherent context for response generation. Experiments on LongMemEval and LoCoMo show that SGMem outperforms strong baselines in long-term conversational QA.

**Strengths:**

- The paper proposes a practical approach for the QA task with long-term context.
- Through extensive experiments, it demonstrates the effectiveness of the proposed method.

**Weaknesses:**

- The approach appears practical, but its novelty is uncertain. Building a graph-based memory from fine-grained information (even if not at the sentence level) for long-term context has been explored in prior work.

- It’s unclear why the graph is built from a single sentence instead of all sentences within a dialogue turn. Providing a justification for this design choice would be beneficial.

- The proposed method effectively scores every sentence from past dialogues, thereby shifting the computational burden from the LLM’s input context to the external database — reducing input size but increasing storage and retrieval overhead. This raises doubts about its practicality and applicability at scale.

- The working mechanism of the proposed approach is described somewhat abstractly, making it difficult to clearly understand how it operates in specific examples. For instance, it’s not clear how the graph’s design choices help the model better understand the context. Demonstrating this with a working example would be beneficial.

- Providing evidence or analysis to illustrate the benefits of sentence-level information would strengthen the paper.

**Questions:**

- How is the context of turn, round, and session implemented? And what does it mean to use each of them (turn, round, and session) as context?

- Could a dialogue turn, rather than a single sentence, be used for constructing the graph memory? If not, what would be the difference between the two approaches?

- The paper states that using only raw dialogue units (turns, rounds, sessions) provides a faithful but fragmented context — is there any evidence supporting this claim?

- Is there any reason only F is always included in the combination? Are there any results for ST, SM, SI, SMI, STM, or STI?

---

### Official Review · Reviewer_wfc3 · 2025-10-31

**Soundness:** 2
**Presentation:** 3
**Contribution:** 2
**Rating:** 2
**Confidence:** 4

**Summary:**

This paper proposes a new conversation memory management mechanism called SGMem. Unlike existing methods, SGMem captures dialogue as a sentence-level graph, allowing for more precise memory alignment than turn level, round level, or session level approaches. SGMem manages not only sentence-level dialogue history but also generated memory, enabling LLMs to produce contextually coherent responses. The authors evaluate SGMem on two benchmark datasets (LongMemEval and LoCoMo) and show that it consistently outperforms existing baselines.

**Strengths:**

1. SGMem adopts sentences as the fundamental memory unit, providing a more fine-grained representation than turns, rounds, or sessions. This allows it to capture contextual dependencies.
2. Semantic relationships between sentences are connected through kNN, linking related sentences (memory units) to maintain contextual coherence across dialogue.
3. SGMem is lightweight, constructed using simple tools such as NLTK rather than relying on LLMs for extraction or graph building.
4. Extensive experiments on the LongMemEval and LoCoMo benchmarks validate the effectiveness of SGMem across diverse baselines, demonstrating its performance in long-term conversation tasks.

**Weaknesses:**

1. There is a lack of examples. The paper needs to show examples like how an actual dialogue session is modeled as a graph and how sentences and memories are connected.
2. In real-world scenarios, since SGMem does not include explicit delete or update operations, the graph will continue to grow indefinitely. This point raises concerns about whether retrieval can remain effective and efficient as the graph becomes excessively large.
3. Using Sentence-BERT and kNN for graph construction is advantageous due to its light weight. However, it may connect semantically irrelevant sentences or miss subtle contextual relations. A case study or error analysis on such cases would strengthen the paper.
4. There is no direct qualitative or quantitative evaluation of the generated memories. The quality of these memories could significantly impact the model's overall performance, but this is not assessed.
5. The proposed SGMem seems highly sensitive to hyperparameters, making it difficult to apply universally. The authors state that optimal settings are difference by dataset, which implies that performance on a new dataset could be unstable depending on hyperparameter. This suggests the method may lack robust generalization.
6. To better demonstrate the robustness of SGMem's sentence-level graph representation, a more extensive ablation study is needed. In particular, comparing sentence-level SGMem with variants built at coarser granularities, such as turn, or round level graphs would help verify whether performance degrades in the absence of sentence-level structuring.

**Questions:**

1. What is the average or typical size of the graph (e.g., number of nodes and edges) constructed by SGMem at the end of a dialogue episode (or session) for each benchmark dataset?
2. The lack of a mechanism to update or delete memories means the graph grows continuously with the conversation. Can this approach be considered efficient in terms of computation and memory? Is it necessary to store everything?
3. Dialogues often include trivial sentences (e.g., greetings, filler words, etc.) that may not require memorization. It is unclear why SGMem chooses to include every sentence as a node rather than filtering for salient or important utterances.
4. I am particularly concerned about whether SGMem would maintain its performance if it were not constructed at the sentence level. Since this design choice is important to the paper's main contribution.
5. The evaluation relies mainly on the LLM-as-a-Judge, which may not fully capture human perceptions of conversation quality. I would encourage the authors to include a human evaluation to complement the automatic metrics. It would be valuable to see whether human judges actually prefer SGMem's memories, and whether they perceive the sentence-graph structure as intuitive or overly complex. These human evaluation could offer deeper insight into how well SGMem aligns with real-world conversational expectations.

---

### Official Review · Reviewer_R5ui · 2025-10-31

**Soundness:** 2
**Presentation:** 3
**Contribution:** 1
**Rating:** 2
**Confidence:** 5

**Summary:**

**Problem & Motivation:**
The paper addresses "memory fragmentation," an issue in long-term conversational agents that occurs as dialogue history grows and exceeds an LLM's context window. This forces the use of a memory system, which typically stores information in two disconnected forms: (1) raw dialogue history, segmented into coarse-grained chunks (e.g., turns, rounds, or full sessions), and (2) generated memory, which are LLM-produced compressed snippets (e.g., summaries, extracted facts, insights). The open problem this paper defines as fragmentation is the dispersal of related information across (1) and (2), and within them. For example, a user's preference might be mentioned in a sentence in Session 1, referenced in Session 5, and summarized in a "fact" snippet. Existing systems may fail to retrieve these three pieces of information coherently, leading to incomplete context and inaccurate LLM responses.


**Related Work & Open Problems:**
The submission's methodology is motivated by the identified failures of existing approaches:
- Chunk-based RAG: This approach, which retrieves raw dialogue chunks (like full sessions), is critiqued for its "coarse granularity." The paper notes that retrieving an entire session to find a single relevant sentence is imprecise and fills the LLM's context with irrelevant noise.
- Entity-graph RAG: This approach tries to be more precise by building a knowledge graph of entities and relations. The paper argues this fails due to two main issues: (1) It is computationally costly, requiring LLM-based extraction for every memory update. (2) It discards rich contextual information by abstracting a full sentence into nodes (e.g., (User) -[likes]- (Dogs)), which the authors argue exacerbates fragmentation by divorcing the "fact" from its original conversational nuance.


**Method:**
The proposed method, SGMem (Sentence Graph Memory), is a framework designed to address the stated fragmentation problem. Its essence is the central design choice of using the sentence as the fundamental unit of memory. The hypothesis is that sentences are (a) more precise than coarse chunks, (b) retain contextual information unlike abstract entities, and (c) are "lightweight" to process (using NLTK for segmentation, not an LLM).
The framework consists of two main components:
- Construction: The system uses **two databases**. A *vector database* is built to index all memory types (raw chunks, sentences, summaries, facts, insights) for fast semantic search. Concurrently, a *graph database* is built to only store the structural relationships of the raw dialogue; this graph contains nodes for Chunks and Sentences, linked by two edge types:
   - E_chunk-sent (Membership Edges): These edges link coarse-grained "Chunk" nodes (e.g., Session_5) to all the "Sentence" nodes that constitute them. They explicitly map between high-level chunks and their fine-grained content.
   - E_sent-sent (Similarity Edges): These edges are K-Nearest-Neighbor (KNN) edges that link semantically related "Sentence" nodes to each other, regardless of which chunk they are in. They link fragmented information across the entire dialogue history.
   - Note: The generated memories—summaries, facts, insights—are indexed in the vector store but are not added as nodes to this graph structure.


**Usage (Retrieval):** The retrieval process is a 4-step mechanism designed to leverage this hybrid structure:
- Retrieve: A user query performs a vector search across the vector database to get two separate sets of results: (1) a set of "seed" sentences ($C_q$) and (2) the Top-K generated memories, defined as Summaries ($M^⁕$), Facts ($F^⁕$), and Insights ($I^⁕$).
- Expand: The seed sentences ($C_q$)—and only the sentences—are used as entry points into the graph. An $h$-hop traversal (using the E_sent-sent edges) finds an expanded set of related sentences ($C^⁕$), pulling in relevant information that the initial vector search missed. This step is intended to de-fragment information.
- Rank Chunks: This expanded set of sentences ($C^⁕$) is mapped up to its parent "Chunk" nodes (using the E_chunk-sent edges). The Chunks are then ranked based on the density of relevant sentences they contain.
- Aggregate: The final context sent to the LLM is the union of the Top-K retrieved raw dialogue Chunks ($K^⁕$) and the generated memories ($M^⁕, F^⁕, I^⁕$) that were retrieved separately in Step 1.

**Experimental Setup:**
The method is evaluated on the LongMemEval benchmark and a 500-question random sample of the LoCoMo dataset. The evaluation metric is Accuracy, determined by an "LLM-as-a-Judge" (Qwen2.5-32B-Instruct). The core models are all-MiniLM-L6-v2 for embeddings and Qwen2.5-32B-Instruct as the generator.

Results:
SGMem-full (both retrieved and generated context) outperforms tested baselines (with and without retrieval), yielding improvements across diverse query types and generally showing robustness to hyperparameter variations. Ablations on query types and hyperparameter values.

**Strengths:**

- The presentation is generally clear, well-structured, and methodologically transparent.
- The experimental protocol benchmarks SGMem against an extensive suite of baselines (prompts without retrieved context, memory management contributions, chunk- and graph-based RAG pipelines), using two established datasets.
- Abalation and sensitivity analyses.

**Weaknesses:**

- *Novelty.* The idea of constructing graphs with multi-granularity nodes (e.g., sentences, keywords, chunks) connected by structural and semantic edges for processing long documents is far from new and has been widely explored across various tasks, including question answering and document summarization. The reviewer believes that the proposed approach does not introduce sufficiently novel or impactful methodological advances beyond existing graph-based retrieval and memory frameworks, and thus does not meet the bar of innovation expected for an ICLR paper.
- *Evaluation bias.* The use of Qwen2.5-32B-Instruct for both generation and LLM-as-a-judge evaluation is a bad practice and introduces a critical methodological flaw, potentially leading to inflated accuracy scores. The issue is compounded by the absence of human validation.
- *No statistical significance tests.* The paper reports performance gains but no statistical significance testing (e.g., paired t-test, bootstrap resampling) is conducted. Given the small sample sizes (500 questions per dataset) and sometimes marginal improvements, it is unclear whether these gains are statistically reliable or due to random variation. The absence of variance measures (standard deviation or confidence intervals) further limits interpretability. For a paper targeting ICLR, where rigorous empirical validation is expected, this omission reduces the credibility of the claimed improvements.
- *Limited analysis of failure modes and qualitative insights.* The main text largely reports aggregate accuracy numbers. There is little diagnostic error analysis in the main body (see only brief mention in limitations and appendix), and it remains unclear how SGMem deals with nuances like ambiguous, underspecified, or contradictory dialogue—especially when generated memory (summaries, facts, insights) itself can be noisy, hallucinated, or factually inconsistent.
- *Reliance on LLM-generated summaries/facts without verification.* While the paper acknowledges factual inconsistencies may arise from LLM outputs (see Limitations), it fails to quantify or systematically analyze the potential negative impacts on retrieval quality or downstream QA accuracy. There is no attempt to mitigate or filter hallucinations.
- *Scope of datasets and external validity.* Both datasets are simulated or curated, and—per the authors’ own admission—lack evaluation for multilingual, multimodal, streaming, or real-world “in-the-wild” conversational memory requirements. Claims about scalability and generalization are thus limited. For example, LoCoMo is long but synthetic; there’s no user study or stress test with truly massive histories, distribution drift, or privacy-sensitive records.
- *Comparative positioning and granularity justification.* Although the presented approach is empirically superior to chunk/entity alternatives, the paper does not conduct an in-depth comparative study on why sentences specifically are optimal, nor does it analyze whether other fine-grained units (e.g., utterance, phrase, semantic frame) may further improve performance or efficiency. The approach seems somewhat arbitrary in this design dimension.
- *Scalability and computational cost.* SGMem’s construction and dual storage of multiple vector tables and a sentence-level graph could prove costly as interaction histories scale—this is briefly mentioned, but the paper lacks any quantitative profiling (memory usage, storage requirements, retrieval/query latency, or graph maintenance costs) on realistic deployment footprints.
- No code was provided to the reviewers, and there is no stated plan for public release and license.

**Questions:**

- How does SGMem perform with respect to storage and inference time as conversational histories grow into the hundreds of thousands or millions of sentences? Do the authors have concrete profiling (latency, RAM/disk requirements, graph traversal times) for large-scale interaction logs?
- In Section 5.4 and Figures 5/6, the ablations suggest sensitivity to $h, k, n, \gamma$, but the paper does not discuss how practitioners should tune these in new/unseen domains. Can the authors provide heuristic guidance for deployment?
- Why is $\epsilon=1$ added to the cosine similarity in Section 3.4?

---

### Meta-Review · Area_Chair_iqhN · 2026-01-07

**Summary:**

All three reviewers agree the paper targets an important and practical problem—long-term conversational memory beyond the context window.
However, the consensus recommendation is reject because the work is viewed as falling below the venue’s bar on novelty and empirical rigor. The most decision-driving concerns are:
- limited methodological novelty / weak positioning
- evaluation methodology concerns, especially reliance on LLM-as-a-judge
- insufficient evidence on scalability / practicality, including lack of profiling for storage, graph growth, and retrieval latency as histories scale.

**Reviewer Concerns:**

Addressed by the rebuttal:
- None. The authors did not provide a rebuttal, so no concerns were substantively resolved in discussion.

Still outstanding:
- Novelty / positioning (all reviewers): SGMem’s core idea (fine-grained graph-based memory + semantic links + retrieval expansion) is considered incremental given existing graph-based retrieval/memory frameworks.
- Evaluation validity (R5ui, wfc3): Heavy dependence on LLM-as-a-judge; lack of human evaluation; potential bias when the judge is closely aligned with the generation model family; no direct evaluation of generated-memory quality.
- Scalability / systems practicality (all reviewers): No quantitative profiling of storage footprint, graph maintenance cost, traversal/query latency.
- Clarity / qualitative evidence (wfc3, o6ii): Insufficient worked examples showing how a dialogue is converted to the graph and how retrieval expansion improves context coherence.
- Robustness / hyperparameters (wfc3, R5ui): Sensitivity to hyperparameters with limited guidance for deployment.

**Reviewer Scores:**

Given no rebuttal, scores would likely remain unchanged after discussion:

R5ui: stays at 2 (reject).
wfc3: stays at 2 (reject).
o6ii: stays at 2 (reject).

---

### Decision · Program_Chairs · 2026-01-26

Reject